# Pollen-food allergy syndrome and component sensitization in adolescents: A Japanese population-based study

**Tomoyuki Kiguchi, Kiwako Yamamoto-Hanada** **\*, Mayako Saito-Abe, Miori Sato, Makoto Irahara, Hiroya Ogita, Yoshitsune Miyagi, Yusuke Inuzuka, Kenji Toyokuni, Koji Nishimura, Fumi Ishikawa, Yumiko Miyaji, Shigenori Kabashima, Tatsuki Fukuie, Masami Narita, Yukihiro Ohya**

Allergy Center, National Center for Child Health and Development, Tokyo, Japan

* yamamoto-k@ncchd.go.jp

## Abstract

Allergic rhino-conjunctivitis with pollen allergy has been prevalent worldwide and Pollen-food allergy syndrome (PFAS) refers to individuals with pollen allergy who develop oral allergy syndrome (OAS) on consuming fruits and vegetables. The prevalence of PFAS varies by region and that in Japanese adolescents remains to be elucidated. In this cross-sectional study, we examined the epidemiological characteristics of PFAS in a general population of Japanese adolescents according to pollen allergy, OAS, and IgE component sensitization. Participants comprised adolescents, at age 13 years, from a prospective birth cohort study in Japan. We administered questionnaires to collect information from parents regarding pollen allergy, PFAS and OAS at each child's age 13 years. ImmunoCAP ISAC was used to assess IgE component sensitization. Among 506 participants with a complete questionnaire and ISAC measurement results, 56.5% had a history of hay fever, 16.0% had a history of OAS, 51.0% had pollen allergy, and 11.7% had a history of PFAS; additionally, 72.7% were sensitized to one or more tree, grass, and/or weed allergens. The most common sensitization (95.7%) among adolescents with pollen allergy was to Japanese cedar (Cry j 1). The most common causal foods were kiwi and pineapple (both 39.0%). Knowledge levels about PFAS were poor among affected adolescents. We found a high prevalence of PFAS among adolescents in Japan. Although it affects approximately 1/10 adolescents in the general population, public awareness regarding PFAS is poor. Interventional strategies are needed to increase knowledge and to prevent PFAS in the general population.

## Introduction

Oral allergy syndrome (OAS) was defined by Amlot *et al.* [1] in 1987 as immediate allergic symptom of the oral mucosa owing to food antigens. Thereafter, the term pollen-food allergy syndrome (PFAS) has been used in patients with pollen allergy who develop OAS after eating fruits and vegetables [2]. PFAS is caused by cross-reactivity between pollen allergens and fruit

**Data Availability Statement:** The data used in the study may not be made public due to ethical considerations and the privacy laws of Japan. The

Act on the Protection of Personal Information (Act No. 57 of May 30, 2003, amended on September 9, 2015) prohibits publication of data containing personal information. All inquiries about access to data should be sent to the Research Office of Allergy Center, National Center for Child Health and Development, Tokyo, Japan, allergy_research@ncchd.go.jp.

**Funding:** This study was funded and supported by the Grant of National Center for Child Health and Development [grant number 26-18].The funders had no role in study design, data collection and analysis, decision to publish, or preparation of the manuscript.

**Competing interests:** The authors have declared that no competing interests exist.

and/or vegetable allergens [3]. Immunoglobulin E (IgE) sensitization to the allergen is required before an allergic reaction to that allergen occurs. Most allergens are proteins, and the protein molecules to which a specific IgE binds are called allergen components. Plant-related allergen components of fruits and vegetables include lipid transfer proteins, profilin, and PR-10 proteins. Because of the structural similarities between allergen components in plants, cross-reactivity can occur in the presence of antibodies that recognize both allergens [4].

Although it has been reported that the prevalence of food allergy and hospital admissions owing to food-induced anaphylaxis in children has been increasing in recent years in the United States and worldwide [5, 6], the trend in the prevalence of PFAS is unclear. As there are regional differences in the trees, grass, and weeds that cause pollen allergy, a recent review by Carlson *et al.* [3] stated that the prevalence of PFAS may vary by region. Although there are several epidemiological reports on PFAS in Japan, some reports were based on hospital data of adult patients [7–9]. As for OAS in Japanese children, a cross-sectional study showed that the prevalence of OAS for Rosaceae fruits and soybean was 0.99% among children in the general population [10]. That study evaluated OAS symptoms for only Rosaceae and soybean, regardless of the presence of pollen allergy. Therefore, the complete picture of the prevalence of PFAS in the general population remains unclear. To the best of our knowledge, there are no epidemiological studies that have evaluated the prevalence of PFAS confirmed by rhinitis, OAS, pollen allergy, and IgE sensitization in the general population of Japan.

The prevalence of allergic rhinoconjunctivitis varies widely from country to country, ranging from 1.4% to 39.7% among adolescents age 13–14 years [11], and pollen allergy is considered common among children. The prevalence of rhinitis is also common; our previous study demonstrated that the prevalence of rhinitis at age 5 years increased rapidly from approximately 11% to approximately 31% at age 9 years [12]. As allergic rhinitis, including pollen allergy, is likely to be further increased, it is speculated that the prevalence of PFAS might also increase in correlation with the increased prevalence of allergic rhinitis and pollen allergy worldwide.

In this study, we sought to examine the epidemiological signature of PFAS confirmed by pollen allergy history and IgE component sensitization among a general population of adolescents.

## Materials and methods

### Study design, setting, and participants

We performed a cross-sectional study of adolescents aged 13 years within a prospective birth cohort study of the general population in the Tokyo Children's Health, Illness and Development Study (T-Child Study) [12–16]. The T-Child Study was conducted at the National Center for Child Health and Development (NCCHD), National Children's Hospital in Tokyo, Japan. A total of 1701 pregnant women who attended their first antenatal visit at the NCCHD were registered from 2003 to 2005. A total of 1550 new-borns were enrolled in the study between March 2004 to August 2006, and the children were followed until age 13 years.

### Ethics statement

This study was conducted in accordance with the principles laid out in the Declaration of Helsinki and other national regulations and guidelines. The T-Child Study was carried out with the approval of the NCCHD Research Ethics Committee (Approval number: 52) and complies with the Japanese ethical guidelines (MHLW) for medical research on humans and the Helsinki Declaration. Written informed consent was obtained from all parents at recruitment and before the questionnaire survey and blood test on behalf of their child. Written informed

assent was obtained from all adolescent participants before the questionnaire survey and blood test.

## Outcome variables

The outcomes evaluated in this study included PFAS, pollen allergy, OAS, and IgE component sensitization. These outcomes and the other variables are defined in Table 1. Clinical symptoms of OAS were evaluated based on questionnaire items, at the child's age 13 years (Table 2).

**Table 1. Definitions of outcomes.**

| Outcome | Definition |
| --- | --- |
| **Rhinitis ever** | A positive answer from the caregiver to the question (child at 13y old), "Has your child ever had a problem with sneezing, or a runny, or blocked nose when he/she DID NOT have a cold or the flu?" |
| **Rhinitis current** | A positive answer from the caregiver to the question (child at 13 y old), "In the past 12 months, has your child had a problem with sneezing, or a runny, or blocked nose when he/she DID NOT have a cold or the flu?" |
| **Current conjunctivitis** | A positive answer from the caregiver to the question (child at 13 y old), "In the past 12 months, has this nose problem been accompanied by itchy-watery eyes?" |
| **Hay fever** | A positive answer from the caregiver to the question (child at 13y old), "Has your child ever had hayfever?" |
| **OAS** | A positive answer from the caregiver to the question (child at 13y old), "Has your child ever had an itchy mouth or redness around his/her mouth after eating fruits and vegetables?" |
| **IgE component sensitization** | An Specific IgE $\geq 0.3$ ISU for either component was considered positive. Serum levels of total and specific IgE measured using ImmunoCAP ISAC were collected. Specific IgE antibody levels were measured within the range of 0.3–100 ISU-E (ISAC standardized units). |
| **Sensitization of trees, grass, or weeds** | Positive for any tree, grass, or weed component in ImmunoCAP ISAC. Included in ImmunoCAP ISAC are trees (Japanese cedar, Cypress, European white birch, alder, London plane tree, hazel, olive), grass, or weeds (Timothy, Bermuda grass, Lamb's quarters, English plantain, Annual mercury, Mugwort, Short ragweed, Pellitory of the wall, Saltwort). |
| **Pollen allergy** | "Hay fever ever at 13y" and "sensitization of trees, grass, or weeds" |
| **PFAS** | "Pollen allergy at 13y" and "OAS at 13y" |
| **Animal allergy ever** | A positive answer from the caregiver to the question (child at 13y old), "Has your child ever experience allergic reactions such as hives, itching eyes, sneezing/nose bleeding, coughing, etc. by touching or approaching animals or entering the room where they were?" |
| **Wheeze ever** | A positive answer from the caregiver to the question (child at 13y old), "Has your child ever had wheezing or whistling in the chest at any time in the past?" |
| **Wheeze current** | A positive answer from the caregiver to the question (child at 13 y old), "Has your child ever had wheezing or whistling in the past 12 months?" |
| **Asthma ever** | A positive answer from the caregiver to the question (child at 13 y old), "Has your child ever had asthma?" |
| **Asthma current** | A positive answer from the caregiver to the question (child at 13 y old), "Has your child ever been diagnosed by a doctor as having asthma in the past 12 months?" |
| **Atopic dermatitis ever** | A positive answer from the caregiver to the question (child at 13 y old), "Has your child ever had atopy (atopic dermatitis)?" |
| **Atopic dermatitis current** | A positive answer from the caregiver to the question (child at 13 y old), "Is your child currently diagnosed with atopic dermatitis by your doctor?" |
| **Recent immediate symptoms of food** | A positive answer from the caregiver to the question (child at 13 y old), "Has your child ever had symptoms such as urticaria and anaphylaxis (strong allergic reaction) after eating certain foods during the last 12 months?" |

OAS, oral allergy syndrome; PFAS, pollen food allergy syndrome.

**Table 2. Questionnaire items about oral allergy syndrome for fruits and vegetables.**

| |
| --- |
| A questionnaire was given to caregivers to evaluate OAS for children and fruits at 13 years of age. |
| The survey included the following questions: |
| 1) Has your child ever had an itchy mouth or redness around his/her mouth after eating fruits and vegetables? Those who chose "Yes" in 1) answered after 2). |
| 2) What did your child eat (multiple answers allowed)? |
| Melons, watermelons, cucumbers, tomatoes, pineapples, kiwis, bananas, avocados, grapes, mangos, apples, peaches, pears, plums, cherries, mandarins, soybeans, soymilk, tofu, edamame, celery, carrots, and others. |
| 3) What were the symptoms (multiple answers allowed)? |
| Itching in the mouth and throat, lips, swelling of eyelids, redness/urticaria (face), redness/urticaria (body/limbs), cough/zeize, vomiting/abdominal pain/diarrhea, and others. |
| 4) Dose your child eat the above fruits and vegetables raw (without heating) in his/her daily life? |
| • He/she took without any particular restrictions. |
| • He/she took some restrictions. |
| • He/she has completely removed it and therefore did not take it. |
| • He/she hasn't had a chance to take it in the last 12 months. |
| 5) Dose the symptoms disappear if the above fruits and vegetables are heated (boiled, baked, canned, jam, ketchup, etc.)? |
| • Yes |
| • No |
| • unknown |

OAS, oral allergy syndrome.

The questionnaires were developed by several certified allergists and epidemiologists. Rhinitis was assessed using the International Study of Asthma and Allergies in Childhood (ISAAC) questionnaire [17–19]. IgE component sensitization was analyzed using ImmunoCAP ISAC (see Table 1).

## Questionnaire survey

A paper-based questionnaire in Japanese, which included the ISAAC and clinical history of OAS and PFAS, was completed by participants' parents. The responses were used to evaluate the health and daily life of adolescents at age 13 years.

## Blood sampling and IgE component measurement

Venous blood samples were obtained from 13-year-old adolescents. Allergen component-specific IgE antibody titres were measured using a multiplex array ImmunoCAP ISAC [20–22] by a private contract laboratory (Thermo Fisher Scientific, Tokyo, Japan). The ImmunoCAP ISAC enables measurement of IgE titres using a fixed panel of the 112 most relevant allergen components from 51 sources in a single test. Specimen management was conducted by a private contract laboratory (SRL, Inc., Tokyo, Japan).

## Bias and study size

Participants were adolescents age 13 years who joined medical check-up of the T-Child Study comprising children of general population. All participated adolescents have been followed since their birth, before they developed PFAS. Study size was inevitably determined by the number of the participants of this cohort study who joined medical check up and answered the questionnaire. They represent the general population of Tokyo metropolitan area in Japan.

## Statistical analysis

The target population in this study was children born as singletons and followed until they reached age 13 years and who had no missing variables for blood tests. Descriptive statistics were performed for all outcomes. The missing values were not imputed. Statistics were performed using JMP version 15 (SAS Institute, Inc., Cary, NC, USA).

## Results

### Participant characteristics

In this study, 726 children age 13 years and their parents responded to the questionnaire, and 506 children age 13 years underwent blood tests and had a completed questionnaire survey.

Table 3 demonstrates the baseline characteristics of participants in this study. Of the 506 children who had undergone blood test (ISAC measurement), 286 (56.5%) had a past history of hay fever, 81 (16.0%) had a history of OAS, 258 (51.0%) had pollen allergy, and 59 (11.7%) had a history of PFAS. In addition, 368 (72.7%) 13-year-olds were sensitized to one or more tree, grass, and/or weed allergens, according to ImmunoCAP ISAC.

Table 4 shows the months in which children had symptoms of rhinitis, within a period of 1 year. Rhinitis was most frequently observed in the following order: March (80.0%), April (67.8%), February (57.1%), and May (40.5%). Rhinitis was most common in the spring season (88.5%), followed by winter (63.5%), autumn (43.5%), and summer (26.9%).

Table 5 shows the sensitization to plant allergens among 258 children who met the definition of having pollen allergy. Japanese cedar (Cry j 1, 95.7%), cypress (Cup a 1, 86.0%), white birch (Bet v 1, 36.0%; Be tv 2, 10.5%; Bet v 4, 1.5%), alder (Aln g 1, 28.3%), short ragweed (Amb a 1, 27.9%), and timothy (Phl p 1, 9.0%; Phl p 11, 2.7%; Phlp12, 6.2%; Phlp2, 2.3%; Phl p 4, 24.0%; Phl p 5, 12.8%; Phl p 6, 8.1%; Phl p 7, 1.9%) were most frequently associated with sensitization. Cry j 1 sensitization was high in both the PFAS and pollen allergy without OAS groups (Cry j1 93.2% vs. 96.5%). The PR-10 protein Bet v 1 was higher in the PFAS group compared with the pollen allergy without OAS group (59.3 vs. 29.1%). This tendency was the same for Aln g 1 (47.5 vs. 22.6%), Cora 1.0101 (45.8 vs. 23.1%), and Cora 1.0401 (50.8 vs. 24.1%). As for the four profilin allergens (Bet v 2, Hev b 8, Mer a 1, and Phl p 12), the PFAS group had a higher percentage of sensitization compared with the pollen allergy without OAS group.

**Table 3. Baseline characteristics of participants in this study.**

| Child characteristics | Questionnaire at 13 years old | | Blood sampling at 13 years old | |
|---|---|---|---|---|
| | N | n(%) | N | n(%) |
| Female sex | 726 | 369(50.8%) | 506 | 256(50.6%) |
| Hay fever | 725 | 406(56.0%) | 506 | 286(56.5%) |
| OAS | 725 | 115(15.9%) | 505 | 81(16.0%) |
| Pollen allergy | - | - | 506 | 258(51.0%) |
| PFAS | - | - | 506 | 59(11.7%) |
| Allergy sensitization | - | - | 506 | 414(81.8%) |
| Sensitization of trees, grass, or weeds | - | - | 506 | 368(72.7%) |
| Income <4,000,000 yen/year | 684 | 62(9.1%) | 475 | 43(9.1%) |
| Maternal allergic rhinitis | 726 | 367(50.6%) | 506 | 274(54.2%) |

N, number of participants; OAS, oral allergy syndrome; PFAS, pollen-food allergy syndrome.

**Table 4. Seasons of the children showing symptoms of rhinitis at 13 years old.**

| Time of the year | Pollen allergy | OAS | PFAS | Current rhinitis |
|---|---|---|---|---|
| | N = 232 | N = 94 | N = 56 | N = 469 |
| **Month** | | | | |
| January | 65(28.0%) | 23(24.5%) | 10(17.9%) | 142(30.3%) |
| February | 146(62.9%) | 49(52.1%) | 29(51.8%) | 268(57.1%) |
| March | 211(90.9%) | 76(80.9%) | 48(85.7%) | 375(80.0%) |
| April | 186(80.2%) | 66(70.2%) | 45(80.4%) | 318(67.8%) |
| May | 89(38.4%) | 38(40.4%) | 25(44.6%) | 190(40.5%) |
| June | 38(16.4%) | 16(17.0%) | 9(16.1%) | 98(20.9%) |
| July | 35(15.1%) | 15(16.0%) | 10(17.9%) | 83(17.7%) |
| August | 33(14.2%) | 17(18.1%) | 9(16.1%) | 77(16.4%) |
| September | 66(28.4%) | 29(30.9%) | 16(28.6%) | 133(28.4%) |
| October | 75(32.3%) | 30(31.9%) | 19(33.9%) | 152(32.4%) |
| November | 67(28.9%) | 24(25.5%) | 14(25.0%) | 138(29.4%) |
| December | 47(20.3%) | 22(23.4%) | 9(16.1%) | 125(26.7%) |
| **Season** | | | | |
| Spring | 221(95.3%) | 85(90.4%) | 53(94.6%) | 415(88.5%) |
| Summer | 53(22.8%) | 25(26.6%) | 15(26.8%) | 126(26.9%) |
| Autumn | 100(43.1%) | 41(43.6%) | 22(39.3%) | 204(43.5%) |
| Winter | 155(66.8%) | 57(60.6%) | 31(55.4%) | 298(63.5%) |

N, number of participants; OAS, oral allergy syndrome; PFAS, pollen-food allergy syndrome.

**Table 5. All-component sensitization by allergic outcomes at 13 years old.**

| Component | Allergen | Current rhinitis | Hay fever | OAS | Pollen allergy | PFAS | Pollen allergy but without OAS |
|---|---|---|---|---|---|---|---|
| | | N = 344 | N = 286 | N = 81 | N = 258 | N = 59 | N = 199 |
| Act d 1 | Kiwi fruit | 6(1.7%) | 5(1.7%) | 1(1.2%) | 4(1.6%) | 1(1.7%) | 3(1.5%) |
| Act d 2 | Kiwi fruit | 11(3.2%) | 8(2.8%) | 6(7.4%) | 8(3.1%) | 5(8.5%) | 3(1.5%) |
| Act d 5 | Kiwi fruit | 0(0%) | 0(0%) | 0(0%) | 0(0%) | 0(0%) | 0(0%) |
| Act d 8 | Kiwi fruit | 26(7.6%) | 25(8.7%) | 12(14.8%) | 25(9.7%) | 12(20.3%) | 13(6.5%) |
| Aln g 1 | Alder | 78(22.7%) | 73(25.5%) | 32(39.5%) | 73(28.3%) | 28(47.5%) | 45(22.6%) |
| Amb a 1 | Short ragweed | 85(24.7%) | 72(25.2%) | 20(24.7%) | 72(27.9%) | 19(32.2%) | 53(26.6%) |
| Ana o 2 | Cashew | 1(0.3%) | 1(0.3%) | 1(1.2%) | 1(0.4%) | 1(1.7%) | 0(0%) |
| Api g 1 | Celery | 27(7.8%) | 25(8.7%) | 13(16.0%) | 25(9.7%) | 12(20.3%) | 13(6.5%) |
| Ara h 1 | Peanut | 4(1.2%) | 5(1.7%) | 3(3.7%) | 5(1.9%) | 3(5.1%) | 2(1.0%) |
| Ara h 2 | Peanut | 6(1.7%) | 7(2.4%) | 3(3.7%) | 7(2.7%) | 3(5.1%) | 4(2.0%) |
| Ara h 3 | Peanut | 1(0.3%) | 2(0.7%) | 2(2.5%) | 2(0.8%) | 2(3.4%) | 0(0%) |
| Ara h 6 | Peanut | 6(1.7%) | 7(2.4%) | 4(4.9%) | 7(2.7%) | 4(6.8%) | 3(1.5%) |
| Ara h 8 | Peanut | 58(16.9%) | 56(19.6%) | 25(30.9%) | 56(21.7%) | 22(37.3%) | 34(17.1%) |
| Ara h 9 | Peanut | 1(0.3%) | 1(0.3%) | 0(0%) | 0(0%) | 0(0%) | 0(0%) |
| Art v 1 | Mugwort | 4(1.2%) | 3(1.0%) | 2(2.5%) | 3(1.2%) | 2(3.4%) | 1(0.5%) |
| Art v 3 | Mugwort | 1(0.3%) | 1(0.3%) | 1(1.2%) | 1(0.4%) | 1(1.7%) | 0(0%) |
| Ber e 1 | Brazil nut | 1(0.3%) | 1(0.3%) | 1(1.2%) | 1(0.4%) | 1(1.7%) | 0(0%) |
| Bet v 1 | Birch | 102(29.7%) | 93(32.5%) | 39(48.1%) | 93(36.0%) | 35(59.3%) | 58(29.1%) |
| Bet v 2 | Birch | 31(9.0%) | 27(9.4%) | 12(14.8%) | 27(10.4%) | 10(16.9%) | 17(8.5%) |
| Bet v 4 | Birch | 6(1.7%) | 4(1.4%) | 2(2.5%) | 4(1.6%) | 2(3.4%) | 2(1.0%) |
| Che a 1 | Lamb's quarters | 3(0.9%) | 3(1.0%) | 1(1.2%) | 3(1.2%) | 1(1.7%) | 2(1.0%) |

*(Continued)*

**Table 5.** (Continued)

| Component | Allergen | Current rhinitis | Hay fever | OAS | Pollen allergy | PFAS | Pollen allergy but without OAS |
|---|---|---|---|---|---|---|---|
| | | N = 344 | N = 286 | N = 81 | N = 258 | N = 59 | N = 199 |
| Cor a 1.0101 | Hazelnut | 78(22.7%) | 73(25.5%) | 31(38.3%) | 73(28.3%) | 27(45.8%) | 46(23.1%) |
| Cor a 1.0401 | Hazelnut | 85(24.7%) | 78(27.3%) | 34(42.0%) | 78(30.2%) | 30(50.8%) | 48(24.1%) |
| Cor a 8 | Hazelnut | 0(0%) | 0(0%) | 0(0%) | 0(0%) | 0(0%) | 0(0%) |
| Cor a 9 | Hazelnut | 2(0.6%) | 2(0.7%) | 2(2.5%) | 2(0.8%) | 2(3.4) | 0(0%) |
| Cry j 1 | Japanese cedar | 270(78.5%) | 247(86.4%) | 66(81.5%) | 247(95.7%) | 55(93.2%) | 192(96.5%) |
| Cup a 1 | Cypress | 238(69.2%) | 222(77.6%) | 55(67.9%) | 222(86.0%) | 48(81.4%) | 174(87.4%) |
| Cyn d 1 | Bermuda grass | 65(18.9%) | 58(20.3%) | 15(18.5%) | 58(22.5%) | 12(20.3%) | 46(23.1%) |
| Gly m 4 | Soybean | 60(17.4%) | 56(19.6%) | 29(35.8) | 56(21.7%) | 24(40.7%) | 32(16.1%) |
| Gly m 5 | Soybean | 0(0%) | 0(0%) | 0(0%) | 0(0%) | 0(0%) | 0(0%) |
| Gly m 6 | Soybean | 1(0.3%) | 1(0.3%) | 1(1.2%) | 1(0.4%) | 1(1.7%) | 0(0%) |
| Hev b 1 | Latex | 0(0%) | 0(0%) | 0(0%) | 0(0%) | 0(0%) | 0(0%) |
| Hev b 3 | Latex | 0(0%) | 0(0%) | 0(0%) | 0(0%) | 0(0%) | 0(0%) |
| Hev b 5 | Latex | 0(0%) | 0(0%) | 0(0%) | 0(0%) | 0(0%) | 0(0%) |
| Hev b 6.01 | Latex | 0(0%) | 0(0%) | 0(0%) | 0(0%) | 0(0%) | 0(0%) |
| Hev b 8 | Latex | 41(11.9%) | 36(12.6%) | 15(18.5%) | 36(14.0%) | 13(22.0%) | 23(11.6%) |
| Jug r 1 | Walnut | 6(1.7%) | 5(1.7%) | 4(4.9%) | 5(1.9%) | 2(3.4%) | 3(1.5%) |
| Jug r 2 | Walnut | 46(13.4%) | 47(16.4%) | 16(19.8%) | 47(18.2%) | 13(22.0%) | 34(17.1%) |
| Jug r 3 | Walnut | 3(0.9%) | 3(1.0%) | 2(2.5%) | 3(1.2%) | 2(3.4%) | 1(0.5%) |
| Mal d 1 | Apple | 90(26.2%) | 85(29.7%) | 37(45.7%) | 85(32.9%) | 33(55.9%) | 52(26.1%) |
| Mer a 1 | Annual mercury | 39(11.3%) | 34(11.9%) | 15(18.5%) | 34(13.2%) | 13(22.0%) | 21(10.6%) |
| Ole e 1 | Olive | 0(0%) | 0(0%) | 0(0%) | 0(0%) | 0(0%) | 0(0%) |
| Ole e 7 | Olive | 0(0%) | 0(0%) | 0(0%) | 0(0%) | 0(0%) | 0(0%) |
| Ole e 9 | Olive | 13(3.8%) | 12(4.2%) | 4(4.9%) | 12(4.7%) | 3(5.1%) | 9(4.5%) |
| Par j 2 | Pellitory of the wall | 3(0.9%) | 2(0.7%) | 1(1.2%) | 2(0.8%) | 1(1.7%) | 1(0.5%) |
| Phl p 1 | Timothy | 59(17.2%) | 49(17.1%) | 17(21.0%) | 49(19.0%) | 13(22.0%) | 36(18.1%) |
| Phl p 11 | Timothy | 7(2.0%) | 7(2.4%) | 4(4.9%) | 7(2.7%) | 2(3.4%) | 5(2.5%) |
| Phl p 12 | Timothy | 19(5.5%) | 16(5.6%) | 10(12.3%) | 16(6.2%) | 8(13.6%) | 8(4.0%) |
| Phl p 2 | Timothy | 5(1.5%) | 6(2.1%) | 4(4.9%) | 6(2.3%) | 3(5.1%) | 3(1.5%) |
| Phl p 4 | Timothy | 66(19.2%) | 62(21.7%) | 22(27.2%) | 62(24.0%) | 18(30.5%) | 44(22.1%) |
| Phl p 5 | Timothy | 33(9.6%) | 33(11.5%) | 13(16.0%) | 33(12.8%) | 10(16.9%) | 23(11.6%) |
| Phl p 6 | Timothy | 21(6.1%) | 21(7.3%) | 9(11.1%) | 21(8.1%) | 6(10.2%) | 15(7.5%) |
| Phl p 7 | Timothy | 7(2.0%) | 5(1.7%) | 3(3.7%) | 5(1.9%) | 3(5.1%) | 2(1.0%) |
| Pla a 1 | London plane | 0(0%) | 0(0%) | 0(0%) | 0(0%) | 0(0%) | 0(0%) |
| Pla a 2 | London plane | 65(18.9%) | 59(20.6%) | 17(21.0%) | 59(22.9%) | 15(25.4%) | 44(22.1%) |
| Pla a 3 | London plane | 3(0.9%) | 3(1.0%) | 2(2.5%) | 3(1.2%) | 2(3.4%) | 1(0.5%) |
| Pla l 1 | English plantain | 1(0.3%) | 1(0.3%) | 1(1.2%) | 1(0.4%) | 1(1.7%) | 0(0%) |
| Pru p 1 | Peach | 80(23.3%) | 74(25.9%) | 37(45.7%) | 74(28.7%) | 32(54.2%) | 42(21.1%) |
| Pru p 3 | Peach | 3(0.9%) | 3(1.0%) | 1(1.2%) | 3(1.2%) | 1(1.7%) | 2(1.0%) |
| Sal k 1 | Saltwort | 1(0.3%) | 0(0%) | 1(1.2%) | 0(0%) | 0(0%) | 0(0%) |
| Ses i 1 | Sesame | 1(0.3%) | 1(0.3%) | 1(1.2%) | 1(0.4%) | 1(1.7%) | 0(0%) |
| Der f 1 | American house dust mite | 240(69.8%) | 198(69.2%) | 55(67.9%) | 186(72.1%) | 45(76.3%) | 141(70.9%) |
| Der f 2 | American house dust mite | 213(61.9%) | 170(59.4%) | 50(61.7%) | 160(62.1%) | 39(66.1%) | 121(60.8%) |
| Der p 1 | European house dust mite | 228(66.3%) | 186(65.0%) | 50(61.7%) | 175(67.8%) | 41(69.5%) | 134(67.3%) |
| Der p 10 | European house dust mite | 9(2.6%) | 4(1.4%) | 2(2.5%) | 4(1.6%) | 1(1.7%) | 3(1.5%) |
| Der p 2 | European house dust mite | 210(61.0%) | 169(59.1%) | 47(58.0%) | 159(61.6%) | 38(64.4%) | 121(60.8%) |
| Can f 1 | Domestic dog | 51(14.8%) | 42(14.7%) | 11(13.6%) | 41(15.9%) | 8(13.6%) | 33(16.6%) |

*(Continued)*

**Table 5.** (Continued)

| Component | Allergen | Current rhinitis | Hay fever | OAS | Pollen allergy | PFAS | Pollen allergy but without OAS |
|---|---|---|---|---|---|---|---|
| | | N = 344 | N = 286 | N = 81 | N = 258 | N = 59 | N = 199 |
| **Can f 2** | Domestic dog | 7(2.0%) | 8(2.8%) | 1(1.2%) | 8(3.1%) | 0(0%) | 8(4.0%) |
| **Can f 3** | Domestic dog | 6(1.7%) | 7(2.4%) | 2(2.5%) | 7(2.7%) | 2(3.4%) | 5(2.5%) |
| **Can f 5** | Domestic dog | 9(2.6%) | 11(3.8%) | 4(4.9%) | 11(4.3%) | 4(6.8%) | 7(3.5%) |
| **Fel d 1** | Domestic cat | 138(40.1%) | 124(43.4%) | 35(43.2%) | 123(47.7%) | 31(52.5%) | 92(46.2%) |
| **Fel d 2** | Domestic cat | 5(1.5%) | 6(2.1%) | 2(2.5%) | 6(2.3%) | 2(3.4%) | 4(2.0%) |
| **Fel d 4** | Domestic cat | 18(5.2%) | 15(5.2%) | 6(7.4%) | 15(5.8%) | 5(8.5%) | 10(5.0%) |

N, number of participants; OAS, oral allergy syndrome; PFAS, pollen-food allergy syndrome.

## Characteristics of patients with PFAS

The characteristics of patients with PFAS are shown in Table 6. Of 59 adolescents with PFAS, 18 (30.5%) reported ever having atopic dermatitis, 21 (35.6%) ever had asthma, and 10 (16.9%) had recent immediate symptoms related to foods.

## Symptoms in patients with PFAS

Clinical symptoms of adolescents with PFAS are shown in Table 7. The most common symptom was itching in the mouth and throat (83.1%); this was followed by swelling of the lips and

**Table 6.** Characteristics of allergic outcomes of PFAS adolescents at 13 years old.

| Allergic outcomes | PFAS | |
|---|---|---|
| | N | n(%) |
| **Child allergy outcomes** | | |
| **Rhinitis ever** | 59 | 57(96.6%) |
| **Rhinitis current** | 59 | 56(94.9%) |
| **Current conjunctivitis** | 59 | 50(84.7%) |
| **Hay fever** | 59 | 59(100%) |
| **OAS** | 59 | 59(100%) |
| **Wheeze ever** | 59 | 27(45.8%) |
| **Wheeze current** | 59 | 6(10.1%) |
| **Asthma ever** | 59 | 21(35.6%) |
| **Asthma current** | 59 | 4(6.8%) |
| **Atopic dermatitis ever** | 59 | 18(30.5%) |
| **Atopic dermatitis current** | 59 | 10(16.9%) |
| **Recent immediate symptoms of food** | 59 | 10(16.9%) |
| **Animal allergy ever** | 59 | 23(39.0%) |
| **Allergy sensitization** | 59 | 59(100%) |
| **Sensitization of trees, grass, or weeds** | 59 | 59(100%) |
| **Environmental exposure** | | |
| **Environmental tobacco smoke** | 58 | 13(22.4%) |
| **Income <4,000,000 yen/year** | 57 | 2(3.5%) |
| **Maternal characteristics** | | |
| **Maternal allergic rhinitis** | 59 | 37(62.7%) |

N, number of participants; PFAS, pollen-food allergy syndrome; OAS, oral allergy syndrome.

**Table 7. Symptoms and causal food allergens in adolescents with PFAS.**

| Characteristic | PFAS |
|---|---|
| | N = 59 |
| **Symptoms** | |
| Discomfort in the mouth and throat | 49 (83.1%) |
| Swelling of the lips and eyelids | 9 (15.3%) |
| Face redness and urticaria | 9 (15.3%) |
| Body redness and urticarial | 6 (10.2%) |
| Cough and wheezing | 2 (3.4%) |
| Vomiting, abdominal pain, and diarrhea | 1 (1.7%) |
| Other symptoms | 7 (11.9%) |
| Anaphylaxis | 0 (0%) |
| **Daily intake of causal raw foods** | |
| Without limitation | 6 (10.2%) |
| With some restrictions | 28 (47.5%) |
| Completely eliminated | 13 (22.0%) |
| Not consumed during the past 12 months | 12 (20.3%) |
| **Symptoms disappear with heating and processing causal foods** | |
| Yes | 21 (35.6%) |
| No | 3 (5.1%) |
| Unknown | 35 (59.3%) |

N, Number of participants; PFAS, pollen-food allergy syndrome

eyelids (15.3%), face redness/urticaria (15.3%), body redness/urticaria (10.2%), cough/wheezing (3.4%), and vomiting/abdominal pain/diarrhea (1.7%). No participants had a history of anaphylaxis. Only 22.0% of adolescents with PFAS had completely eliminated the causal food(s) from their diet. Heating and processing causal foods led to the elimination of symptoms in 35.6% of children with PFAS. Adolescents with PFAS who were not aware whether their symptoms disappeared when eating heated or processed causal foods accounted for the largest proportion (59.3%) of the study population.

## Causal foods of PFAS

Causal foods of PFAS are shown in Table 8. The most common causal foods were kiwi fruit and pineapple (39.0%), followed by peach (28.8%), apple (22.0%), tomato (18.6%), melon (16.9%), mango (13.6%), cherry (11.9%), watermelon (8.5%), and pear (6.8%).

**Table 8. Causal food allergen according to OAS and PFAS.**

| Allergen details | PFAS |
|---|---|
| | N = 59 |
| **Causal food allergen** | |
| Kiwi fruit | 23(39.0%) |
| Pineapple | 23(39.0%) |
| Peach | 17(28.8%) |
| Apple | 13(22.0%) |
| Tomato | 11(18.6%) |
| Melon | 10(16.9%) |

*(Continued)*

**Table 8.** (Continued)

| Allergen details | PFAS |
|---|---|
| | **N = 59** |
| **Watermelon** | 5(8.5%) |
| **Mango** | 8(13.6%) |
| **Cherry** | 7(11.9%) |
| **Pear** | 4(6.8%) |
| **Yam** | 3(5.1%) |
| **Avocado** | 3(5.1%) |
| **Banana** | 3(5.1%) |
| **Loquat** | 3(5.1%) |
| **Soybean** | 3(5.1%) |
| **Cucumber** | 1(1.7%) |
| **Celery** | 1(1.7%) |
| **Carrot** | 1(1.7%) |
| **Plum** | 2(3.4%) |
| **Grapes** | 1(1.7%) |
| **Orange** | 0(0%) |
| **Okura** | 0(0%) |
| **Blueberry** | 1(1.7%) |
| **Hazelnut** | 1(1.7%) |
| **Lotus root** | 0(0%) |
| **Potato** | 1(1.7%) |
| **Corn** | 1(1.7%) |
| **Eggplant** | 1(1.7%) |
| **Yuzu** | 1(1.7%) |
| **Lychee** | 1(1.7%) |
| **Peanuts** | 1(1.7%) |
| **Fig** | 0(0%) |
| **Spinach** | 0(0%) |
| **Grapefruit** | 0(0%) |
| **Number of causal food allergen** | |
| **1** | 22(37.3%) |
| **2** | 18(30.5%) |
| **3** | 5(8.5%) |
| **4** | 7(11.9%) |
| **5** | 2(3.4%) |
| **6** | 2(3.4%) |
| **7** | 2(3.4%) |
| **10** | 1(1.7%) |

N, number of participants; PFAS, pollen-food allergy syndrome.

As for the number of causal foods, 22 adolescents reported only one food, followed by 2 (30.5%), 3 (8.5%), 4 (11.9%), 5 (3.4%), 6 (3.4%), 7 (3.4%), and 10 foods (1.7%).

## Sensitization status of PFAS

We examined association between causal foods and sensitization status. The most common allergen associated with sensitization among adolescents with in PFAS in this study was Cry j 1 (93.2%), shown in Table 4). ImmunoCAP ISAC includes four components (Cora1.0101,

Cora1.0401, Betv1, Alng1) as PR-10 proteins of trees/grass/weeds, and four components (Betv2, Hevb8, Mera1, Phlp12) as profilin of trees/grass/weeds. The sensitization status of PR-10 protein, profilin, and Cry j 1 of trees, grass, and weeds for each food causing PFAS are shown in Table 9.

## Discussion

To our best knowledge, this was the first report of the epidemiological signatures of PFAS confirmed by a clinical history of pollen allergy and OAS and component sensitization, using data of a general population of adolescents from a birth cohort in Japan. This study revealed a high prevalence of pollen allergy and PFAS among adolescents. Common causal foods of PFAS were kiwi, pineapple, peach, and apple. A past history of allergy such as atopic dermatitis and asthma was not very common among adolescents with PFAS. The high prevalence of PFAS in adolescents found in this study revealed the possibility that PFAS is becoming more common in adolescents than previously thought.

### Pollen allergy

Japanese cedar is widely distributed throughout Japan, including in Tokyo. A questionnaire survey by Okuda *et al.* [23] showed that the prevalence of hay fever caused by Japanese cedar was approximately 20% in children age 10–19 years across Japan. The present study results coincided with those previous findings, as the most common sensitization among adolescents with pollen allergy was to Japanese cedar.

Birch is a common tree found in parts of Hokkaido and Nagano, but not in Tokyo. However, alder, which is a type of birch tree, is common in parks of Tokyo. At least 36% of adolescents with pollen allergy showed Bet v 1 sensitization. A review by Biedermann *et al.* [24] found that birch pollen sensitization ranged from 8% to 16% in the general population of Europe. From our previous study, Bet v 1 sensitization was identified in 13.9% of a general population of children age 9 years [12]. The population of Japan might have similar rates of sensitization to birch and alder as those in Europe.

In our present study, we observed relatively few participants with sensitization to grasses and weeds, such as ragweed and timothy grass. We speculate that there were few grasses and weeds near participants' residential area as the adolescents included in this study lived around Tokyo.

### PFAS

In this study, we revealed that 11.7% of adolescents in our study had PFAS, and 22.9% with pollen allergy had PFAS. Unfortunately, the actual prevalence of PFAS among children outside of Japan remains unclear as there are no reports regarding PFAS in these populations. From case series, the prevalence of PFAS among study participants with pollen allergy was 33.6% in one European report [25], 41.7% in a Korean study [26] including children and adults, and 12.1% among Australian children [27]. As mentioned earlier, PFAS prevalence differs according to region.

In our study, PFAS accounted for 22.9% of adolescents with pollen allergy. The percentage of PFAS among our participants with pollen allergy was lower than that of the Korean population. The reason may be that Japanese cedar is the most common allergen associated with pollen allergy in Japan. According to age-independent reports from Japan, the prevalence of PFAS in Japanese patients with cedar pollinosis is approximately 10%–13% [9, 23]. Furthermore, Cry j 1 sensitization was most common among trees, grasses, and weeds in adolescents with pollen allergy in our study, as compared with a report from South Korea where birch and

**Table 9. Causal foods and sensitization (PR-10, profilin, and Cry j 1) in PFAS patients.**

| Component type | Component name | Allergen | Kiwi fruit (n = 23) | Pineapple (n = 23) | Peach (n = 17) | Apple (n = 13) | Tomato (n = 11) | Melon (n = 10) | Mango (n = 8) | Cherry (n = 7) | Watermelon (n = 5) | Pear (n = 4) | Yam (n = 3) | Avocado (n = 3) | Banana (n = 3) | Loquat (n = 3) | Soybean (n = 3) | Plum (n = 2) |
|---|---|---|---|---|---|---|---|---|---|---|---|---|---|---|---|---|---|---|
| Pectate-lyase | Cry j 1 | Japanese cedar | 21 (91.3%) | 20 (87.0%) | 16 (94.1%) | 13 (100%) | 10 (90.9%) | 9 (90.0%) | 7 (87.5%) | 6 (85.7%) | 4 (80.0%) | 4 (100%) | 3 (100%) | 3 (100%) | 3 (100%) | 2 (66.7%) | 3 (100%) | 2 (100%) |
| Profilin | Mer a 1 | Annual mercury | 6 (26.1%) | 4 (17.4%) | 3 (17.6%) | 4 (30.8%) | 4 (36.4%) | 2 (20.0%) | 3 (37.5%) | 1 (14.3%) | 1 (20.0%) | 0 (0%) | 1 (33.3%) | 3 (100%) | 0 (0%) | 1 (33.3%) | 1 (33.3%) | 1 (50.0%) |
| Profilin | Phl p 12 | Timothy | 5 (21.7%) | 3 (13.0%) | 2 (11.8%) | 1 (7.7%) | 4 (36.4%) | 2 (20.0%) | 2 (25.0%) | 0 (0%) | 1 (20.0%) | 0 (0%) | 1 (33.3%) | 2 (66.7%) | 0 (0%) | 1 (33.3%) | 0 (0%) | 0 (0%) |
| Profilin | Hev b 8 | Latex | 6 (26.1%) | 5 (21.7%) | 3 (17.6%) | 3 (23.1%) | 5 (45.5%) | 2 (20.0%) | 3 (37.5%) | 1 (14.3%) | 2 (40.0%) | 0 (0%) | 1 (33.3%) | 3 (100%) | 0 (0%) | 1 (33.3%) | 1 (33.3%) | 1 (50.0%) |
| Profilin | Bet v 2 | Birch | 5 (21.7%) | 3 (13.0%) | 3 (17.6%) | 2 (15.4%) | 4 (36.4%) | 2 (20.0%) | 2 (25.0%) | 1 (14.3%) | 1 (20.0%) | 0 (0%) | 1 (33.3%) | 2 (66.7%) | 0 (0%) | 1 (33.3%) | 1 (33.3%) | 1 (50.0%) |
| PR-10 | Act d 8 | Kiwi fruit | 7 (30.4%) | 2 (8.7%) | 7 (41.2%) | 5 (38.5%) | 1 (9.1%) | 2 (20.0%) | 1 (12.5%) | 3 (42.9%) | 0 (0%) | 1 (25.0%) | 0 (0%) | 1 (33.3%) | 1 (33.3%) | 0 (0%) | 2 (66.7%) | 1 (50.0%) |
| PR-10 | Api g 1 | Celery | 8 (34.8%) | 4 (17.4%) | 5 (29.4%) | 4 (30.8%) | 3 (27.3%) | 3 (30.0%) | 2 (25.0%) | 1 (14.3%) | 0 (0%) | 0 (0%) | 1 (33.3%) | 2 (66.7%) | 0 (0%) | 0 (0%) | 2 (66.7%) | 1 (50.0%) |
| PR-10 | Gly m 4 | Soybean | 11 (47.8%) | 8 (34.8%) | 11 (64.7%) | 11 (84.6%) | 4 (36.4%) | 4 (40.0%) | 3 (37.5%) | 6 (85.7%) | 0 (0%) | 4 (100%) | 1 (33.3%) | 2 (66.7%) | 1 (33.3%) | 0 (0%) | 2 (66.7%) | 2 (100%) |
| PR-10 | Ara h 8 | Peanut | 9 (39.1%) | 7 (30.4%) | 10 (58.8%) | 8 (61.5%) | 4 (36.4%) | 5 (50.0%) | 5 (62.5%) | 4 (57.1%) | 0 (0%) | 2 (50.0%) | 1 (33.3%) | 2 (66.7%) | 2 (66.7%) | 1 (33.3%) | 2 (66.7%) | 2 (100%) |
| PR-10 | Pru p 1 | Peach | 14 (60.9%) | 10 (43.5%) | 14 (82.4%) | 11 (84.6%) | 6 (54.5%) | 6 (60.0%) | 6 (75.0%) | 7 (100%) | 3 (60.0%) | 4 (100%) | 2 (66.7%) | 2 (66.7%) | 3 (100%) | 3 (100%) | 2 (66.7%) | 2 (100%) |
| PR-10 | Mal d 1 | Apple | 14 (60.9%) | 9 (39.1%) | 13 (76.5%) | 12 (92.3%) | 5 (45.5%) | 6 (60.0%) | 5 (62.5%) | 6 (85.7%) | 2 (40.0%) | 4 (100%) | 2 (66.7%) | 2 (66.7%) | 3 (100%) | 2 (66.7%) | 2 (66.7%) | 2 (100%) |
| PR-10 | Cor a 1.0401 | Hazelnut | 13 (56.5%) | 9 (39.1%) | 14 (82.4%) | 12 (92.3%) | 6 (54.5%) | 6 (60.0%) | 6 (75.0%) | 7 (100%) | 2 (40.0%) | 4 (100%) | 2 (66.7%) | 2 (66.7%) | 3 (100%) | 2 (66.7%) | 2 (66.7%) | 2 (100%) |
| PR-10 | Cor a 1.0101 | Hazelnut | 11 (47.8%) | 9 (39.1%) | 13 (76.5%) | 10 (76.9%) | 4 (36.4%) | 5 (50.0%) | 5 (62.5%) | 6 (85.7%) | 1 (20.0%) | 3 (75.0%) | 2 (66.7%) | 2 (66.7%) | 3 (100%) | 1 (33.3%) | 2 (66.7%) | 1 (50.0%) |
| PR-10 | Aln g 1 | Alder | 12 (52.2%) | 8 (34.8%) | 13 (76.5%) | 12 (92.3%) | 4 (36.4%) | 5 (50.0%) | 5 (62.5%) | 6 (85.7%) | 0 (0%) | 4 (100%) | 2 (66.7%) | 2 (66.7%) | 3 (100%) | 1 (33.3%) | 2 (66.7%) | 1 (50.0%) |
| PR-10 | Bet v 1 | Birch | 14 (60.9%) | 10 (43.5%) | 15 (88.2%) | 12 (92.3%) | 7 (63.6%) | 6 (60.0%) | 6 (75.0%) | 7 (100%) | 4 (80.0%) | 4 (100%) | 2 (66.7%) | 2 (66.7%) | 3 (100%) | 3 (100%) | 2 (66.7%) | 2 (100%) |

The sensitization status (Bet v 1, Aln g 1, and Bet v 2) of PR-10 protein and profilin in trees/grasses/weeds for causative foods of the family Rosaceae was as follows peach (Bet v 1, 88.2%; Aln g 1, 76.5%; Bet v 2, 17.6%), pear (Bet v 1, 100%; Aln g 1, 100%; Bet v 2, 0%), apple (Bet v 1, 92.3%; Aln g 1, 92.3%; Bet v 2, 15.4%), and cherries (Bet v 1, 100%; Aln g 1, 85.7%; Bet v 2, 14.3%). Adolescents with PFAS owing to tomato showed 90.9% sensitization to Cry j 1.

alder sensitization were most common [26]. As the prevalence of pollen allergy caused by Japanese cedar is highest in Japan, this study showed that the prevalence of PFAS in pollen allergy is relatively low.

According to our study findings, 59.3% of adolescents with PFAS did not know whether their symptoms would disappear if they ate causal foods that were heated or processed. There are several possible reasons for this result. First, awareness among the general public about PFAS may still be low. In other words, most Japanese people have no knowledge that heating and processing causal foods can reduce the symptoms of PFAS. Second, eating habits among Japanese people involve less processing and cooking at home compared with populations in Europe and the United States. Nonetheless, as PFAS becomes more common, public awareness about how to manage PFAS should be improved.

## Sensitization status of PFAS in adolescents

The most common allergen resulting in sensitization was Cry j 1 (93.2%) among adolescents with PFAS in this study. Regarding the PR-10 protein in foods, sensitization to the PR-10 protein in apples and peaches was highly positive in those with PFAS symptoms after consuming apples or peaches, which was consistent with a past report by Shirasaki *et al.* [28] in Japan. However, for PFAS symptoms caused by kiwi, the positive rate of sensitization to the PR-10 protein related to kiwi was not higher than that of apple and peach. It is suggested that the degree of involvement of the PR-10 protein varies depending on the food that causes PFAS.

In this study, participants who had PFAS symptoms with foods in the Rosaceae family (e.g., apples, peaches, pears, cherries, plums) were more sensitized to the birch family PR-10 protein. This result is consistent with a past report of Bet v 1 cross-reactivity with allergens of the Rosaceae family [29].

Using a questionnaire survey, Osawa *et al.* [30] demonstrated that melons, kiwis, and pineapples were the causative foods of OAS symptoms among Japanese children age 10 years and older. In this study, the top causative foods in PFAS were kiwi and pineapple, and a high percentage of sensitization to Japanese cedar (Cry j 1) was observed. We speculate that certain individuals who have OAS symptoms with pineapple and kiwi may be affected by cross-reactivity with Japanese cedar pollinosis. However, another theory has been raised, namely, it is also possible that some adolescents who complain of OAS symptoms with pineapple or kiwi may have non-allergic mechanisms owing to proteolytic enzymes such as bromelain and actinidin.

In this study, few people complained of PFAS symptoms with tomatoes, despite the high percentage of Cry j 1 sensitization observed. Cry j 2 has been pointed out as a cedar allergen that is largely involved in tomato PFAS [31]. Therefore, we considered the possibility that adolescents with PFAS might be less sensitized to Cry j 2. It has been pointed out in a previous report that because Cry j 1 is abundant on the outer wall surface of Japanese cedar pollen and the outer layer of orbicles, the moulting phenomenon is not always an essential condition when acting as an allergen, but Cry j 2 is released as an allergen only after Japanese cedar pollen has molted. Sensitization patterns might differ between Cry j 1 and Cry j 2 [32].

## Limitations

This study has some limitations. First, although the background was similar between the 726 participants with completed questionnaires and the 506 participants who also had ISAC measurements, ISAC measurement could not be performed in all participants at age 13 years. In addition, information about a history of rhinitis and history of OAS with fruits and vegetables was based on questionnaire responses given by children's caregivers and not a doctor's

diagnosis. From the above, it is possible that the prevalence of pollen allergy and PFAS was underestimated or overestimated. Second, because this survey was conducted in Tokyo, the area of residence of participating children was limited. As has been pointed out, regional differences exist in pollen allergy; therefore, further investigation of regional differences in PFAS is necessary. Unfortunately, this study may make it difficult to take a closer look at the association between allergens and the season of the rhinitis symptoms because the various pollen allergens in each season in Japan (e.g., Japanese cedar, cypress, birch, and alder in spring) commonly lead to multiple allergen IgE sensitizations. Therefore, it is impossible to specify the specific pollen allergen for the pollen allergy in each child.

## Conclusions

Our study findings revealed a high prevalence of PFAS among adolescents in Tokyo metropolitan area in Japan. These results suggest that PFAS is common in adolescents; however, most adolescents with PFAS in this study had poor knowledge about PFAS. Early intervention with respect to allergenic foods can prevent immediate food allergy to peanut and hen's egg, although interventional strategies against PFAS remains to be developed. The issue of PFAS has been neglected, and further investigation is needed to explore interventional strategies against the global PFAS epidemic. Furthermore, public awareness about PFAS should be encouraged.

## Acknowledgments

We deeply thank the mothers and children who participated in the T-CHLD Study.

We also thank Analisa Avila, ELS, of Edanz Group (https://en-author-services.edanzgroup. com/ac) and Abigail, editor, of ENAGO (https://www.enago.com/) for editing a draft of this manuscript.

## Author Contributions

**Conceptualization:** Tomoyuki Kiguchi, Kiwako Yamamoto-Hanada, Yukihiro Ohya.

**Data curation:** Tomoyuki Kiguchi.

**Formal analysis:** Tomoyuki Kiguchi.

**Funding acquisition:** Yukihiro Ohya.

**Investigation:** Kiwako Yamamoto-Hanada, Mayako Saito-Abe, Miori Sato, Makoto Irahara, Hiroya Ogita, Yoshitsune Miyagi, Yusuke Inuzuka, Kenji Toyokuni, Koji Nishimura, Fumi Ishikawa, Yumiko Miyaji, Shigenori Kabashima, Tatsuki Fukuie, Masami Narita, Yukihiro Ohya.

**Methodology:** Tomoyuki Kiguchi, Kiwako Yamamoto-Hanada, Mayako Saito-Abe, Miori Sato, Makoto Irahara, Hiroya Ogita, Yoshitsune Miyagi, Yusuke Inuzuka, Kenji Toyokuni, Koji Nishimura, Fumi Ishikawa, Yumiko Miyaji, Shigenori Kabashima, Tatsuki Fukuie, Masami Narita, Yukihiro Ohya.

**Project administration:** Kiwako Yamamoto-Hanada, Masami Narita, Yukihiro Ohya.

**Supervision:** Kiwako Yamamoto-Hanada, Yumiko Miyaji, Tatsuki Fukuie, Yukihiro Ohya.

**Writing – original draft:** Tomoyuki Kiguchi.

**Writing – review & editing:** Kiwako Yamamoto-Hanada, Mayako Saito-Abe, Miori Sato, Makoto Irahara, Hiroya Ogita, Yoshitsune Miyagi, Yusuke Inuzuka, Kenji Toyokuni, Koji

Nishimura, Fumi Ishikawa, Yumiko Miyaji, Shigenori Kabashima, Tatsuki Fukuie, Masami Narita, Yukihiro Ohya.

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
