## [Decision Letter · Decision Letter 0]

21 Jan 2021

PONE-D-20-40156

Pollen-food allergy syndrome and component sensitization in adolescents: a Japanese population-based study

PLOS ONE

Dear Dr. Yamamoto‐Hanada,

Thank you for submitting your manuscript to PLOS ONE. After careful consideration, we feel that it has merit but does not fully meet PLOS ONE’s publication criteria as it currently stands. Therefore, we invite you to submit a revised version of the manuscript that addresses the points raised during the review process.

We look forward to receiving your revised manuscript.

Kind regards,

Linglin Xie

Academic Editor

PLOS ONE

Journal Requirements:

Reviewers' comments:

Reviewer's Responses to Questions

**Comments to the Author**

1. Is the manuscript technically sound, and do the data support the conclusions?

Reviewer #1: Yes

Reviewer #2: Partly

2. Has the statistical analysis been performed appropriately and rigorously? 

Reviewer #1: Yes

Reviewer #2: Yes

3. Have the authors made all data underlying the findings in their manuscript fully available?

Reviewer #1: Yes

Reviewer #2: Yes

4. Is the manuscript presented in an intelligible fashion and written in standard English?

Reviewer #1: Yes

Reviewer #2: Yes

5. Review Comments to the Author

Reviewer #1: 1st Review

Summary:

Kiguchi et al conducted a cross-sectional study using questionnaires on pollen allergy, PFAS, and OAS as well as ImmunoCAP SAC to assess IgE component sensitization among 13-year-old adolescents residing in the Tokyo region. Pollen-food allergy syndrome (PFAS) is termed for oral allergy syndrome that occurs among those with pollen allergy. The authors reported a high prevalence of PFAS than previously thought. Kiwi and pineapple were found to be common causal foods. The authors discussed the possible cross reactivity between cedar allergy with common PFAS causative foods. Despite the high PFAS prevalence, the understanding of PFAS among the participants appear to be low. The authors call for the development of prevention strategies and the education of the public regarding PFAS.

Major concerns:

1. The authors mention that their findings suggest “PFAS is becoming more common in adolescents than previously thought”. However, the paper does not discuss statistical evidence that suggest a change in prevalence. I believe it would be helpful for the reader if the paper discussed relevant statistics that shows a change in prevalence over time.

2. Background information regarding IgE component sensitization would be helpful for those who are unfamiliar to the field of immunology.

Minor concerns:

1. There appears to be some run-on sentences.

2. The current format of Table 9 makes the words and numbers fall into two rows. Will this table be published longitudinally to allow for the words and numbers to fit into one row for easy reading?

Reviewer #2: 1. Na and n^a in the manuscript both represent "Number of participants without missing values". Maybe you could use one of them through the paper. In addition, I feel like the Na or n^a in the paper represents that number of participants who have the symptom or something like that. "Number of participants without missing values" is confusing to me.

2. Is there a connection between the seasons of the children showing symptoms of rhinitis and the allergens?

3. Group1: participants with pollen allergy but without OAS. Group2: participants with PFAS. Could you do more comparisons between the two groups so that we may find more mechanisms about the PFAS and know how to manage it?

6. PLOS authors have the option to publish the peer review history of their article (what does this mean?). If published, this will include your full peer review and any attached files.

Reviewer #1: No

Reviewer #2: No

---

## [Author Response · Author response to Decision Letter 0]

26 Feb 2021

PLOS ONE

PONE-D-20-40156

Pollen-food allergy syndrome and component sensitization in adolescents: a Japanese population-based study

Reviewer #1: 1st Review

Major concerns:

1. The authors mention that their findings suggest “PFAS is becoming more common in adolescents than previously thought”. However, the paper does not discuss statistical evidence that suggest a change in prevalence. I believe it would be helpful for the reader if the paper discussed relevant statistics that shows a change in prevalence over time.

Response: Thank you for your comment. As you have pointed out, this article does not provide epidemiological evidence to suggest changes in the prevalence of pediatric PFAS.

Prevalence data for PFAS in children in Japan are inadequate. Therefore, it is inappropriate to show expressions that refer to changes in the prevalence of PFAS in children over time. We have revised the relevant parts in the Abstract, Discussion, and Conclusion sections accordingly.

2. Background information regarding IgE component sensitization would be helpful for those who are unfamiliar to the field of immunology.

Response: Thank you for your suggestion. We agree with your comment, and we have added an explanation about IgE components in the Introduction section as follows: “Immunoglobulin E (IgE) sensitization to the allergen is required before an allergic reaction to that allergen occurs. Most allergens are proteins, and the protein molecules to which a specific IgE binds are called allergen components. Plant-related allergen components of fruits and vegetables include lipid transfer proteins, profilin, and PR-10 proteins. Because of the structural similarities between allergen components in plants, cross-reactivity can occur in the presence of antibodies that recognize both allergens.”

We also added the following reference:

Matricardi PM, Kleine-Tebbe J, Hoffmann HJ, Valenta R, Hilger C, Hofmaier S, et al. EAACI Molecular Allergology User’s Guide. Pediatr Allergy Immunol. 2016;27 Suppl 23:1-250.

Minor concerns:

1. There appears to be some run-on sentences.

Response: Thank you for the comments. A native English editor has corrected the run-on sentences and explanations.

2. The current format of Table 9 makes the words and numbers fall into two rows. Will this table be published longitudinally to allow for the words and numbers to fit into one row for easy reading?

Response: Thank you for your comment. We have changed the orientation of Table 9 from portrait to landscape in the revised manuscript.

Reviewer #2:

1. Na and n^a in the manuscript both represent "Number of participants without missing values". Maybe you could use one of them through the paper. In addition, I feel like the Na or n^a in the paper represents that number of participants who have the symptom or something like that. "Number of participants without missing values" is confusing to me.

Response: Thank you for pointing this out. We have changed to “Na” and “na” to “N” or “n” throughout the revised manuscript. Furthermore, we have changed the explanation from “number of participants without missing values” to “number of participants.”

2. Is there a connection between the seasons of the children showing symptoms of rhinitis and the allergens?

Response: Thank you for your question. Unfortunately, our study may make it difficult to take a closer look at the association between allergens and the season of the rhinitis symptoms because the various pollen allergens in each season in Japan (e.g., Japanese cedar, cypress, birch, and alder in spring) commonly lead to multiple allergen IgE sensitizations. Therefore, it is impossible to specify the specific pollen allergen for the pollen allergy in each child. We have added this explanation to the Limitations part of the Discussion in the revised manuscript.

3. Group1: participants with pollen allergy but without OAS. Group2: participants with PFAS. Could you do more comparisons between the two groups so that we may find more mechanisms about the PFAS and know how to manage it?

Response: Thank you for your important comment. Because our study is an epidemiological study, not a basic science study, the results are only suggestive of the mechanism of PFAS. However, there were several differences in the types of allergen components that caused sensitization between the pollen allergy, but without the OAS group and the PFAS group. We have added the following explanation to the revised manuscript: “Cry j 1 sensitization was high in both the PFAS and pollen allergy without the OAS groups (Cry j1 93.2% vs. 96.5%). The PR-10 protein Bet v 1 was higher in the PFAS group than the pollen allergy without the OAS group (59.3% vs. 29.1%). This tendency was the same for Aln g 1 (47.5% vs. 22.6%), Cora 1.0101 (45.8% vs. 23.1%), and Cora 1.0401 (50.8% vs. 24.1%). As for the four profilin allergens (Bet v 2, Hev b 8, Mer a 1, and Phl p 12), the PFAS group had a higher percentage of sensitization than the pollen allergy without the OAS group.”

---

## [Decision Letter · Decision Letter 1]

23 Mar 2021

Pollen-food allergy syndrome and component sensitization in adolescents: a Japanese population-based study

PONE-D-20-40156R1

Dear Dr. Yamamoto-Hanada,

We’re pleased to inform you that your manuscript has been judged scientifically suitable for publication and will be formally accepted for publication once it meets all outstanding technical requirements.

Kind regards,

Linglin Xie

Academic Editor

PLOS ONE

Additional Editor Comments (optional):

Reviewers' comments:

Reviewer's Responses to Questions

**Comments to the Author**

1. If the authors have adequately addressed your comments raised in a previous round of review and you feel that this manuscript is now acceptable for publication, you may indicate that here to bypass the “Comments to the Author” section, enter your conflict of interest statement in the “Confidential to Editor” section, and submit your "Accept" recommendation.

Reviewer #1: All comments have been addressed

Reviewer #2: (No Response)

2. Is the manuscript technically sound, and do the data support the conclusions?

Reviewer #1: Yes

Reviewer #2: (No Response)

3. Has the statistical analysis been performed appropriately and rigorously? 

Reviewer #1: Yes

Reviewer #2: (No Response)

4. Have the authors made all data underlying the findings in their manuscript fully available?

Reviewer #1: Yes

Reviewer #2: (No Response)

5. Is the manuscript presented in an intelligible fashion and written in standard English?

Reviewer #1: Yes

Reviewer #2: (No Response)

6. Review Comments to the Author

Reviewer #1: The authors addressed the questions and suggestions effectively. The additional information regarding IgE sensitization and the limitations of the study is well discussed. Statements about the PFAS statistics among adolescents now reflect the current data. The tables are also easier to read now. Overall, the authors effectively addressed the questions and suggestions making the paper smoother to read and understand.

Reviewer #2: (No Response)

7. PLOS authors have the option to publish the peer review history of their article (what does this mean?). If published, this will include your full peer review and any attached files.

Reviewer #1: No

Reviewer #2: No

---

## [Editor Report · Acceptance letter]

5 Apr 2021

PONE-D-20-40156R1 

Pollen-food allergy syndrome and component sensitization in adolescents: a Japanese population-based study 

Dear Dr. Yamamoto-Hanada:

I'm pleased to inform you that your manuscript has been deemed suitable for publication in PLOS ONE. Congratulations! Your manuscript is now with our production department. 

Kind regards, 

on behalf of

Dr. Linglin Xie 

Academic Editor

PLOS ONE